# The Spectrum and Carrier Frequencies of Common Pathogenic Cystic Fibrosis Transmembrane Conductance Regulator Gene Mutations in Men from the General Population: The Role of Ethnicity

**DOI:** 10.3390/ijms26146625

**Published:** 2025-07-10

**Authors:** Ludmila Osadchuk, Mikhail Ivanov, Elena Komova, Alexander Osadchuk

**Affiliations:** 1Department of Human Molecular Genetics, Federal Research Center ‘Institute of Cytology and Genetics’, The Siberian Branch of the Russian Academy of Sciences, 630090 Novosibirsk, Russia; osadchuk@bionet.nsc.ru; 2Joint Stock Company Vector Best, Novosibirsk District, 630117 Novosibirsk, Russia; ivanovmk@vector-best.ru (M.I.);

**Keywords:** CFTR, spectrum of CFTR variants, carrier frequency, cystic fibrosis, semen quality, reproductive hormones, general population, ethnic differences

## Abstract

Mutations in the CFTR gene, which cause the autosomal recessive disease cystic fibrosis, can also affect male fertility. The aim of this study was to investigate the spectrum and carrier frequency of common pathogenic CFTR variants among men from the general population, analyze ethnic differences, and assess associations with male fertility indicators. Male volunteers (*n* = 1895) from six cities in Russia and Belarus were analyzed for the carrier frequencies of 17 pathogenic CFTR variants and two polymorphisms, as well as semen quality and reproductive hormone levels. Heterozygous carriers of six pathogenic CFTR mutations, F508del, G542X, N1303K, 3849+10kbC>T, CFTRdele2,3, and R117C, and two polymorphisms, IVS9-5T and 5T-(12-13) TG, were identified, with cumulative frequencies of 2.06% and 6.65%, respectively. Significant ethnic differences were revealed in the spectrum and carrier frequencies of pathogenic CFTR variants among Slavs, Buryats, and Yakuts. Slavs exhibited a high proportion of heterozygous carriers of CFTR mutations (2.70%), whereas none were detected among Buryats and Yakuts. The highest carrier frequency for the CFTR polymorphism was observed among Slavs (8.35%), followed by Buryats (5.83%) and Yakuts (1.36%). No association was found between the carriers of identified CFTR variants and male fertility indicators. Thus, the spectrum and carrier frequency of genetic CFTR variants are determined by the ethnic composition of the population, providing a basis for ethnicity-specific screening of pathogenic CFTR variants.

## 1. Introduction

One of the current challenges in reproductive genetics is the search for genetic causes of impaired reproductive potential in men. The genetic background of male infertility and subfertility is complex, with more than 2300 genes involved in spermatogenesis and spermiogenesis, but known genetic factors account for only ~20–25% of male infertility cases [1]. The most common genetic causes of male infertility and subfertility include chromosomal aberrations [2,3], Y-chromosome microdeletions [3,4], mutations and polymorphisms of the androgen receptor gene AR [5], mutations of genes associated with congenital hypogonadotropic hypogonadism and Kallmann syndrome [6,7], and pathogenic variants of the transmembrane conductance regulator gene (CFTR) [8,9,10].

The CFTR gene is located on chromosome 7q31.2, contains 27 exons, and encodes the CFTR protein, which regulates electrolyte and fluid transport in almost all epithelial tissues with exocrine function, including the lung, pancreas, and male reproductive tract. CFTR is a transmembrane protein that forms a selective cAMP-dependent chloride channel present in the apical membrane of epithelial cells lining most exocrine glands. Defects in the CFTR protein synthesis and malfunction of the CFTR protein lead to a common monogenic autosomal recessive disorder, cystic fibrosis (CF), and CFTR-related disorders such as congenital bilateral aplasia of the vas deferens (CBAVD), bronchiectasis, and chronic pancreatitis. In CF patients, glandular cells retain potassium and sodium ions, and the secretions they produce are characterized by excessive viscosity, leading to the obstruction of ducts and disruption of normal organ function [8,9,11].

Different CFTR mutations vary in the degree of their clinical manifestations. The classical form represents a serious disorder characterized by the impaired functions of the exocrine glands and dysfunctions of the respiratory system, gastrointestinal tract, pancreas, and reproductive system. Pathology of the respiratory tract is the primary cause of severe CF complications and mortality [8,12]. Today, significant progress in the diagnosis and therapy of CF patients has resulted in significantly improved patient survival rates, with the median age of survival approaching 50 years [13]. Geographical and ethnic features of CF prevalence have been established; for instance, in most European countries and North America, the prevalence of CF ranges from 1:2000 to 1:4000 newborns [8], whereas Russia exhibits a lower frequency of CF at 1:9000 newborns [14]. CF is rare in Asian and African populations; for example, its prevalence among the Chinese is 1:128,000 [15], and among the Japanese, it is 1:350,000 [16].

To date, more than 2000 different CFTR variants have been identified, and the list of novel CFTR variants is constantly growing; however, only a part of these (1085 genetic variants) has been clearly defined as CF-causing [17]. In Russia, 275 different pathogenic variants of the CFTR gene in CF patients have been registered [14]. The most commonly encountered mutations include F508del, CFTRdele2,3, E92K, 2184insA, 3849+10kbC>T, 2143delT, G542X, N1303K, W1282X, and L138ins. Among these, the F508del mutation, in both homozygous and compound heterozygous forms, is the most widespread, accounting for 50–70% of cases in both Russia and worldwide, although the frequency of the F508del variant among Asian people with CF is only 12–31% compared to the European CF population [8,11,14,15,18,19,20].

According to the clinical manifestations of the disease, CFTR mutations are divided into three groups: (1) those leading to CF, (2) those associated with CFTR-related disorders, and (3) those that do not have clinical manifestations. Functional classification is based on changes in the CFTR protein production process, according to which pathogenic CFTR mutations are traditionally divided into six classes [8,11,12,13,17,21,22]. Classes I-III, categorized as “severe” mutations, lead to a reduction in or absence of the CFTR protein on the apical membrane of cells and impairment of the exocrine function of the pancreas. The remaining IV-VI classes, in which residual chloride channel function is retained, are classified as “mild”. In the case of “severe” mutations, the classical form of CF is observed. However, in some genetic CFTR variants, the chloride channel function is preserved, but the function of CFTR as a bicarbonate channel is altered, which is accompanied by disturbances in the bronchopulmonary and reproductive systems [8,14,21,22]. The phenotypic manifestation of CF varies greatly, ranging from severe to subclinical forms, and depends on genotype, modifier genes, environmental factors, and the treatment intensity, so often, the genotype does not allow for an accurate prediction regarding the severity and prognosis of the disease.

Increased survival rates among patients with CF or CFTR-related disorders undoubtedly make a married couple think about reproductive plans when CF is diagnosed in at least one partner. Men with CF face complex issues related to sexual and reproductive health, including infertility and hypogonadism, but normal spermatogenesis in these patients persists, although it may be weakened [8,9,22,23,24]. Infertility affects more than 95–98% of men with CF, with obstructive azoospermia being the primary cause of infertility. This condition occurs due to CBAVD, which leads to a complete absence of sperm in the ejaculate and prevents natural conception and is considered the genital form of CF [8,11,12]. Most patients with CBAVD can have biological children by obtaining sperm through surgical retrieval of sperm and using ART [23]. Nevertheless, approximately 2% of patients with CF remain fertile; they are typically carriers of “mild” CFTR mutations [9,24]. A higher frequency of CFTR mutations is established in cases of non-obstructive azoospermia and oligozoospermia, suggesting the involvement of the CFTR protein in spermatogenesis or sperm maturation [25], in addition to its role in the development of the epididymis and vas deferens. Recent data indicate that the CFTR protein provides establishment of adequate environment for germ cell development in the testis and maturation in the epididymis [26]. Many men with CBAVD and infertility do not exhibit other clinical signs of CF; however, many of them carry compound heterozygotes with “severe” and “mild” CFTR mutations, and in most cases, the “mild” mutation is represented by the 5T allele [22]. Some mutant variants of the CFTR gene can be observed in healthy men, whose only clinical condition is a reduction in semen quality [22].

Some polymorphisms may affect the transcription or function of the CFTR protein. Polymorphic markers identified in the structure of the CFTR gene are classified as “mild” mutations, for example, a stretch of five thymidines located in intron 9 of the CFTR gene results in the loss of exon 10, as well as the formation of only 10% of the normal protein and causes malformation of the vas deferens [8,27]. The IVS9-5T allele (formerly known as IVS8-5T) was found in approximately 10% of individuals in the general population. The IVS9-5T is considered a “mild” mutation with incomplete penetrance, although its clinical significance remains uncertain. Nevertheless, there are suggestions that the IVS9-5T variant may be associated with an increased risk of non-obstructive male infertility [28]. Splicing disruption often occurs when the 5T allele is in the *cis*-combination with a polymorphism of TG-repeats (alleles 12TG or 13TG), leading to a change in the level of the normal CFTR protein. The *trans*-combination of the 5T allele with “severe” mutation of the CFTR gene, such as F508del or R117H, may be a cause of CBAVD and infertility [27]. It has been proposed that among carriers of 5T alleles, only the *cis*-combinations with the genetic variants 5T-12TG and 5T-13TG might contribute to impaired spermatogenesis in some ethnic populations [27,29].

From a practical point of view, the identification of heterozygous carriers of pathogenic CFTR mutations may help assess the risk of having sick children in a couple planning pregnancy and, ultimately, reduce the CF incidence rate. On the other hand, the increasing number of identified CFTR mutations and the observed variability in their phenotypic manifestations create a basis for studying genotype-phenotype correlations, in particular, genetic associations with male fertility indicators in different ethnic populations. In Russia and Belarus, studies have been conducted on the spectrum and prevalence of CFTR gene mutations, mainly in infertile men with CF or suffering from CBAVD or CUAVD (unilateral absence of the vas deferens) [30,31,32,33]. Data on the spectrum and frequency of heterozygous carriers of pathogenic CFTR variants in the general population in these countries are limited [34,35,36,37]; moreover, to date, systematic population-based studies of the spectrum and frequency of pathogenic CFTR variants for many ethnic groups in Russia and Belarus have not yet been conducted. In addition, the association of heterozygous carriage of many pathogenic mutations or polymorphisms of the CFTR gene with sperm quality and reproductive hormones remains unresolved.

The aim of this study was to estimate the spectrum and carrier frequency of common pathogenic mutations and polymorphisms in the CFTR gene among men from the Russian Federation and the Republic of Belarus—countries that are similar in ethnic, historical, geographical, and cultural aspects. Additionally, the associations of heterozygous carriage of pathogenic CFTR variants with indicators of semen quality and hormonal status were investigated. The research was conducted on men recruited from the general multiethnic population, which included several fairly large ethnic groups.

## 2. Results

### 2.1. Characteristics of the Entire Study Population

The entire study population consisted of 1895 participants, but only Slavs (*n* = 1186), Buryats (*n* = 223), and Yakuts (*n* = 147) represented significant ethnic cohorts, which together accounted for 82.1% of the entire study population. The remaining participants were representatives of other ethnicities or descendants of mixed marriages. The Slavic ethnic groups living in Russia and Belarus did not differ in the frequency of heterozygous carriers of the identified CFTR mutations and polymorphisms (Appendix A). Both Slavic groups were combined for further analysis.

Of the entire study population, 1831 men (96.6%) provided an ejaculate. Azoospermia was observed in 2.0%, pathozoospermia was observed in 39.6%, and normozoospermia was observed in 61.1% of men. In the group of azoospermia, there were no sperm in the ejaculate; in the group of pathozoospermia, the concentration and percentage of progressively motile and morphologically normal sperm were lower than the reference values (either each indicator or any combination thereof), and in the last group of normozoospermia, the sperm concentration was equal to or exceeded 16.0 million/mL, the percentage of progressively motile sperm was equal to or exceeded 30%, and the percentage of sperm with normal morphology was equal to or exceeded 4.0% (the semen quality indicators corresponded to the reference values of the norm according to the WHO recommendations [38,39]).

### 2.2. Spectrum and Carrier Frequency of Identified CFTR Variants in the Entire Study Population and Selected Ethnic Groups

The molecular genetic analysis of 17 CFTR mutations revealed the carriers of six types of pathogenic CFTR variants: F508del, G542X, N1303K, 3849+10kbC>T, CFTRdele2,3, and R117C. Eleven types of the CFTR gene variants have not been identified, including E92X, E92K, R334W, W1282X, L138ins, 2184insA, 2143delT, 1677delTA, S1196X, I507del, and L467F. The frequency of heterozygous carriage of the CFTR mutations among men from the entire study population was 2.06% (Table 1). The most common mutation was F508del (29 carriers), while the number of carriers of the remaining five mutations ranged from 1 to 2. The IVS9-5T polymorphism was identified in 104 carriers (5.49%). The Tn (TG)m polymorphism was found in 21 participants, of which 11 were carriers of 5T-12TG, and 10 were carriers of 5T-13TG. Four carriers of the following genotypes were also found: R117C/5T, F508del/5T, F508del/5T-12TG, and 5T/5T.

A comparative analysis of the spectrum and carrier frequency of identified CFTR mutations was carried out among the three most numerous groups—Slavs, Buryats, and Yakuts—and this allowed us to identify the ethnic specifics (Table 1). In the Slavic group, 32 carriers (2.70%) of the heterozygous CFTR mutations and 90 carriers (7.59%) of the IVS9-5T polymorphism were found. Of the pathogenic CFTR variants detected in Slavs, the most common were the F508del and IVS9-5T. In addition, eight carriers of the 5T-12TG polymorphism and one carrier of the genotypes F508del/5T, F508del/5T-12TG, R117C/5T, and 5T/5T were also detected in Slavs. No heterozygous carriers of pathogenic CFTR mutations were found in the Buryat and Yakut groups. In the Buryat group, 3 carriers of the IVS9-5T and 10 carriers of the 5T-13TG polymorphism were identified. One carrier of the IVS9-5T polymorphism and one carrier of the 5T-12TG polymorphism were identified in the Yakut group.

The carrier frequencies of pathogenic CFTR variants were compared between different ethnic groups using the Yates’ Chi-squared (χ^2^) test. Highly significant differences have been found between ethnic groups in the total carrier frequency of pathogenic CFTR variants (χ^2^ = 23.86; *p* = 0.0001) as well as in the total carrier frequency of heterozygotes (ꭓ^2^ = 11.82; *p* = 0.003) and polymorphisms (χ^2^ = 10.43; *p* = 0.005). Unlike the Slavs, the total carrier frequency of heterozygotes was significantly reduced among Buryats (*p* = 0.018) and Yakuts (*p* = 0.043) due to the primary contribution of the F508del variant to the total frequency of heterozygous carriage. Highly significant differences have been found between three ethnic groups in the total carrier frequency of polymorphisms, primarily due to a reduced frequency among Yakuts (*p* = 0.005). Highly significant differences between ethnic groups were also shown in the frequency of the IVS9-5T polymorphism (χ^2^ = 22.48; *p* = 0.00001), primarily due to a reduced frequency among Buryats (*p* = 0.002) and Yakuts (*p* = 0.004). Additionally, highly significant differences between ethnic groups were shown in the total carrier frequency of 5T-(12-13) TG polymorphism (χ^2^ = 21.47; *p* = 0.00002), mainly due to a reduced frequency among Yakuts (*p* = 0.334).

### 2.3. Search for Associations of Identified Pathogenic CFTR Variants with Semen Quality and Hormonal Status

The relationship between pathogenic CFTR variants and reproductive parameters has been studied in two ways. Firstly, to assess the contribution of the identified CFTR mutations and polymorphisms to impaired spermatogenesis, groups of men with azoospermia, pathozoospermia, and normozoospermia were formed from the entire study population and compared based on the spectrum and carrier frequency of genetic CFTR variants (Table 2).

In the azoospermia group, there was only one carrier of the IVS9-5T variant, the pathozoospermia group included 8 heterozygous carriers of the F508del, 1-G542X, and 1-N1303K mutations, as well as 42 carriers of the IVS9-5T and 7 carriers of the 5T-(12-13) TG polymorphisms (Table 2). The carrier frequencies of pathogenic CFTR variants were compared between different semen quality groups using the Yates’ Chi-squared (χ^2^) test. However, no significant differences in the total carrier frequency of pathogenic CFTR variants (χ^2^ = 3.97; *p* = 0.410) as well as in the carrier frequency of heterozygotes (χ^2^ = 2.57; *p* = 0.277) or polymorphisms (χ^2^ = 1.25; *p* = 0.537) were found between groups with different semen quality, suggesting that the identified CFTR mutations are not associated with an increased risk of impaired spermatogenesis.

The second way consisted of comparing semen and hormonal indicators between carriers and non-carriers of pathogenic CFTR variants in the most numerous ethnic groups (Table 3). It should be noted that sperm quality varies between ethnic groups; in particular, the lowest total sperm count was found among the Yakuts, the highest among the Slavs, and the Buryats were in the middle, as reported earlier [40]. The comparative analysis was carried out on the basis of a one-way ANOVA and the method of correction for multiple comparisons—Duncan’s test. There were no significant differences in any male fertility indicator between the groups with different CFTR variants, as in Slavs and Buryats. In the Yakut group (*n* = 2), the carriers of the IVS9-5T and 5T-12TG alleles exhibited decreased semen parameters (oligoasthenozoospermia) and lower testosterone levels relative to the reference values for the norm [38,39]. However, the small number of carriers of pathogenic CFTR variants did not allow us to draw adequate conclusions, and we limited our comparative ethnic analysis to the Slavs and Buryats. The results obtained confirm that carriers of the identified pathogenic CFTR variants do not have an increased risk of impaired male fertility. It is worth pointing out that the semen and hormonal parameters in the R117C/5T, F508del/5T, F508del/5T-12TG, and 5T/5T genotypes were also within the normal range.

## 3. Discussion

The features of the CFTR mutation spectrum vary by ethnic or racial background; therefore, characterizing the spectrum of CFTR variants across geographic locations and different ethnic groups is important for adequate molecular diagnoses of CF, carrier testing in various populations, as well as for individual care and treatment of people with CF. In a recent study [41], the diversity of CFTR variants across six regions of the world (African, Central South Asia, East Asia, European, American/Admixed American, Middle East) was characterized by 454,787 whole-exome sequences. The highest number of CFTR variants was detected in Europeans (*n* = 3192), while the American group had the least number of CFTR variants (*n* = 151). Across the remaining regions, the number of CFTR variants detected was also much lower than in Europeans. The F508del was the most prevalent CF-causing variant found in all regions, except in East Asia, where V520F was the most prevalent. In Russian CF patients, as well as in carriers of pathogenic CFTR variants, the most common mutation is also F508del [19,20,30,31,32,33,34,35,36,37], including the results of a recent study. It is interesting to note that Europeans had the most unique CFTR variants (*n* = 2212), while the Americans had the least unique variants (*n* = 23) [41]. In CF patients from the international CFTR2 database, the five most common mutations are F508del, G542X, G551D, N1303K, and W1282X in descending order of prevalence [17], a list which is different from the five most common Russian CFTR mutations, F508del, CFTRdele2.3, E92K, 1677delTA, and 3849+10kbC>T [14], including the results of our study.

In Slavic populations, the CFTRdele2.3 can be considered a unique mutation, which is identified in Russia with a frequency of 3.5–7.6% in CF patients and 0.10–0.43% in the general population [14,19,20,30,31,35,36], while its prevalence among European CF patients is much lower at 0.62% [17]. No unique CFTR mutations were found in Chinese CBAVD patients, except for the IVS9-5T allele [23]. In Egyptian CF children, the most commonly encountered CFTR mutations were F508del (58%), followed by 2183AA/G (10%), but unique mutations were R1162X (6%) and A544E (4%), which were not reported elsewhere in the Arab population [42]. Despite limited information on the spectrum and frequency of CFTR mutations in Indian CF patients, a heterogeneous spectrum of CFTR gene variants was found in men with CBAVD, which differed from that in the Caucasian population [43]. The most common mutation was the F508del (8.75%), followed by the non-CF-causing mutation R75Q (4.2%); however, the IVS8-5T variant was registered with a higher frequency (42.5%). In Iraqi CF patients, the most common CFTR variants were 3120+1G>A and W1282X, although the F508del was also identified. Surprisingly, the I507del variant was identified in this study, although it was not reported in other neighboring Arab countries [44]. In Jordanian patients with clinical features of CF, the F508del mutation was the most frequent, followed by the N1303K and G85E [45]. Taken together, these data suggest that the European landscape of the most common CFTR variants may not match other regions of the world, including the Russian and especially Asian communities.

The purpose of CF carrier screening in the general population is to identify the carriers of heterozygous pathogenic CF-causing mutations in people without CF or CF-related disorders, who do not have a family history of CF or a partner with CF. Guided by this purpose, we have analyzed the carrier frequency of 17 pathogenic CFTR variants in 1895 men from the general population. In our study population, only six types of pathogenic CFTR mutations were identified—F508del, 3849+10kbC>T, N1303K, R117C, G542X, and CFTRdele2,3—with 2.06% of the study participants having at least one pathogenic CFTR mutation. Our results can be compared to the results of other population studies in the world. The carrier frequency of pathogenic CFTR variants in Europe has become well defined in recent years due to the screening programs and the improvement in CF patient registries [46]. Similar to the high frequency of clinical CF in Europe, the carrier frequency of pathogenic CFTR variants in Europeans is expectedly to be high and on average equal to 3.3–4.0% [47], which is higher than in our study population. Incidentally, the common pathogenic CFTR dele2,3 variant designated as Slavic has been found in many regions of Europe with pronounced East Slavic origin (Czech Republic, Russia, Belarus, Austria, Germany, Poland, Slovenia, Ukraine, and Slovakia) but is practically absent in the vast majority of the remaining countries [47]. Mutational heterogeneity in estimates of the overall carrier frequency of pathogenic CFTR mutations is observed across European countries. In the Italian general population, the CFTR carrier frequency is 3.23%, but in the group of sperm donors, the CFTR carrier frequency is 17% with the carriage of at least one pathogenic CFTR variant [48,49]. In Portugal, the carrier frequency of pathogenic CFTR variants is 3.3%, and the majority of CFTR variants detected have been associated with a less severe CF phenotype [50]. In the general Danish population, F508del carriers account for 3.0% [51], which is higher than in our study population. These findings suggest a higher rate of carrier frequency of pathogenic CFTR variants in European populations in comparison with Russian populations. In the US, 3.8% of the pan-ethnic population are CF carriers, with one functional copy of CFTR and one copy with one or more pathogenic mutations, which is higher than in our study [52]. The authors examined the common CFTR variants by racial/ethnic groups using NGS and showed that 44.0% of carriers identified (among East Asians, South East Asians, and African Americans) had new variants, which would not be detected by a standard genotyping panel. Therefore, exome and genome sequencing may further expand the spectrum of pathogenic CFTR variants identified in the standard genotyping panels, which is of significant interest for mutation screening in communities with a low CF incidence, such as Buryats or Yakuts in Russia.

Scant information is available regarding the prevalence and population-specific genetic spectrum of the CFTR mutations in China. CF is rare in China, and newborn screening and genetic testing are not used in most provinces in China, resulting in a large number of patients with mild or atypical CF remaining undiagnosed. Analysis of CFTR mutations in Chinese patients with CF demonstrated a different spectrum of CFTR variants compared to the Caucasian population, with the most common variant being the G970D variant [15,53], but the F508del mutation was not predominant. The estimated carrier frequency of CFTR variants in the general population was reported as 0.52–0.59, which is significantly lower than in European countries and in our study, although, given the overall huge population of China, a significant number of CFTR mutation carriers can be assumed.

Our results regarding the carrier frequency of pathogenic CFTR mutations are very similar to the results obtained in other Russian population studies. Analysis of the 29 CFTR mutations among blood donors of both sexes (*n* = 1000) revealed a carrier frequency of 2.9% [34]. In a similar study, the carrier frequency of the 11 most common pathogenic CFTR mutations was 2.12% among residents of the Moscow region (*n* = 2168) [35]. In a population study of residents from various regions of Russia (*n* = 922), based on the analysis of 19 common CFTR gene mutations, the carrier frequency of all pathogenic mutations was 2.82% [31]. Considering that the genotyping methods used in the above studies do not detect the spectrum of rare pathogenic mutations in the CFTR gene, their overall burden in the general population may be slightly higher. For instance, a simultaneous analysis of 60 CFTR gene variants in Russians (*n* = 642) from the general population in the Northwestern District of Russia revealed a proportion of 3.58% heterozygous carriers [19]. Whole-genome sequencing of Russians without CF and malignant neoplasms (*n* = 1825), as well as healthy Russians from the general population (*n* = 10,000), identified the carriage of pathogenic CFTR variants in 2.85% of the first group and 2.21% of the second group [36]. At the same time, the heterozygous carrier frequency of “severe” CFTR mutations estimated in Russian men with impaired fertility and azoospermia was higher than in the general population, reaching 4.70% in infertile men [54] and 4.82% in men with azoospermia [55]. In our multiethnic study population, the most common pathogenic CFTR variant was F508del, detected with a carrier frequency of 1.53%. Similar data were obtained from Russian blood donors (1.5%) [34], residents of the Moscow region (1.3%) [35], and in a population sample from various Russian regions (1.4%) [31] or from the Northwestern District of Russia (2.02%) [19].

Long-time heterozygous carriers of pathogenic CFTR variants were not at increased risk of developing CF or CFTR-related diseases; however, the impact of heterozygous carriage of clinically significant genetic CFTR variants in the development of various types of pathology is no longer as harmless as previously thought. Several studies have shown that heterozygous carriers of CFTR mutations have an increased risk of pancreatitis, male infertility, and the development of respiratory infections such as chronic bronchitis, sinusitis, bronchiectasis, and lung cancer [51,56,57]. Considering that in CF patients, the CFTR protein involves in development of the epididymis and vas deferens, as well as in spermatogenesis, sperm maturation, luteinizing hormone, and testosterone levels [9,23,24,25,26], and also taking into account data about a higher CFTR mutation frequency in individuals with impaired sperm production or hypogonadism [24,25,29,58,59], a search for associations of pathogenic CFTR variants with semen quality and hormonal status was carried out in current study. It turned out that the heterozygous carriers of identified CFTR mutations did not differ from the non-carriers in semen parameters and levels of reproductive hormones, including testosterone. In addition, the carrier frequency of identified CFTR mutations in the groups with azoospermia, pathozoospermia, or normal semen quality did not differ, indicating no significant risk of impaired spermatogenesis in men carrying these CFTR variants. Our results are in accordance with results obtained in Spanish sperm donors, who also did not demonstrate significant differences in semen parameters between carriers and non-carriers of pathogenic CFTR variants [49]. The available data suggest the preservation of normal sperm production in heterozygous carriers of pathogenic CFTR variants and may partially explain their stable prevalence in the general population and, consequently, the persisting CF frequency.

However, maintaining normal sperm production in heterozygous carriers of pathogenic CFTR variants cannot exclude a partial CFTR protein dysfunction, leading to impaired sperm fertility properties. The available data are contradictory. While one study showed that the ability of spermatozoa to capacitate remains the same in heterozygous F508del carriers and healthy sperm donors [60], another study demonstrated that the CFTR protein is involved in spermatozoa capacitation, acrosome reaction, and sperm–oocyte fusion [61]. Moreover, impaired CFTR protein expression in spermatozoa correlated with a reduction in sperm quality. Obviously, further studies are needed to expand our knowledge on the impact of heterozygous carriage of pathogenic CFTR variants on spermatogenesis and sperm fertilizing capacity.

To assess the polymorphic background of the CFTR gene in men from the general population and the prevalence of clinically significant polymorphisms, we conducted a genotypic analysis of the IVS9-5T and 5T-(12-13) TG polymorphisms. The carrier frequency of the “mild” splicing mutation IVS9-5T was 5.49%, and IVS9-5T in combination with 12TG or 13TG (“mild” variants) was 1.11%. In Russian men with reproductive disorders in a couple or infertility, the carriage of the IVS9-5T polymorphism was 4.7% and 5.3%, respectively [54,55], which is close to the values obtained in our study. In the general European population, the carrier frequency of the IVS9-5T polymorphism was about 10% [8,27], which is higher than in other Russian studies, including our study.

Our interest in the IVS9-5T and 5T-(12-13) TG polymorphisms was related to their possible negative impact on spermatogenesis, identified in some studies [27,59,62], although other genetic association studies have not revealed any negative effects [63,64]. In our study, there was no significant difference in the carrier frequency of the IVS9-5T, 5T-12TG, and 5T-13TG between azoospermic, pathozoospermic, and normozoospermic men. We also did not find any significant differences in semen and hormonal parameters between carriers and non-carriers of the IVS9-5T or 5T-(12-13) TG in certain ethnic groups—Slavs and Buryats. Our results contradict the results of some other studies, which reported that IVS9-5T alone or in combination with 12TG may increase the susceptibility risk of non-obstructive azoospermia, especially in non-Europeans [59,62]. However, some other reports also did not reveal any significant difference in the carrier frequency of IVS9-5T between oligozoospermic men and the normal control group [63,64]. The observed high heterogeneity of the results may be due to the different ethnic backgrounds of the participants included in the study or the small size of the research groups.

The most interesting result of the current study is the identification of ethnic features in the spectrum and carrier frequency of common pathogenic CFTR variants. Our study demonstrated that heterozygous carriers of pathogenic CFTR mutations were mainly concentrated among Slavs, while the absence of these carriers was observed among Buryats and Yakuts, although carriers of the 5T polymorphism were found. Consequently, Asian ethnic groups like Buryats and Yakuts may have a different spectrum of CFTR gene mutations compared to Slavs or an extremely low frequency of their carriage. However, the spectrum of CFTR gene variants for these ethnic groups remains unknown due to the very low prevalence of CF [14,65,66]. Indeed, in the Republic of Buryatia and the Republic of Sakha (Yakutia), the CF prevalence is 0.5% and 0.3% of the total CF prevalence in Russia [14]. Apparently, CF has not been detected in Yakuts, since screening of CF in 112,164 newborns in 2006–2013 revealed only four Slavic patients and not a single Yakut [65,66]. The carriage and structure of CFTR mutations in the Buryat population are also not fully understood, since cases of CF or CF-related disorders are considered rare [14]. In a previous Russian study [37], ethnic differences in the spectrum and frequencies of CFTR gene variants were shown in a healthy population (*n* = 5505). In this study, Nogais and Bashkirs (Mongoloid and Turkic origin, respectively) demonstrated a very low carrier frequency of CFTR mutations, and F508del, the most common in European populations, was not detected in these ethnic groups. These data are consistent with our data on Buryats and Yakuts, who have similar ethnic origin (Mongoloid and Turkic origin, respectively).

Our study also provided evidence of ethnic differences in the carrier frequency of IVS9-5T, being the highest among Slavs, then in Buryats and Yakuts. Additionally, 5T-12TG was found only among Slavs and Yakuts, and 5T-13TG was identified exclusively among Buryats. It is noteworthy that the IVS9-5T polymorphism is the most common genetic variant of the CFTR gene among Chinese as opposed to the European population, allowing it to be classified as a “hotspot” mutation [11]. The IVS9-5T variant was found in 45.7% of Chinese patients with CBAVD [23], while the 5T allele was found in 64% of Russian patients with CBAVD [20]. In European populations, patients with CBAVD predominantly exhibit “severe” mutations, primarily F508del, with a frequency of about 28% [11], whereas in Chinese patients with CBAVD, the F508del mutation is extremely rare [23,67]. The ethnic differences established in our and other studies could have significant implications for clinical diagnosis and counseling practices in certain ethnic groups.

In our study, there are a few limitations associated with some methodological approaches. Firstly, the ethnic composition of the participants was established based on questionnaires and information on ancestry up to the third generation. Self-reported ethnicity is used in most epidemiological studies, including the current study, to estimate a person’s ancestry and classify the study population by ethnicity. As the world’s population increasingly does not fit into traditional homogeneous ethnic populations and becomes more mixed, the reliability of information about self-reported ethnicity is becoming more problematic. Using self-reported ethnicity as an indirect indicator of genetic origin in cases of unknown or incorrect information about an individual’s ethnic origin can potentially be a source of false results. In practice, a person’s ethnic origin can be estimated using ancestral information markers (AIMs), which are sets of genetic variations for a particular DNA sequence (autosomal, mtDNA, Y-DNA) that occur at different frequencies in many populations around the world. AIMs are used to estimate the ethnic origin of a person’s ancestors, usually expressed as the proportion of ancestors from different ethnic groups. Using AIMs allows us to compare human polymorphisms at these markers with previously analyzed reference sets of genomes from people whose ancestry is reasonably well known. At present, these genetic markers are not absolutely accurate indicators of ethnicity but rather reflect the genetic history of a population. It is possible that a set of stronger ethno-specific markers will be developed for future genetic ancestry marker validation.

Secondly, an important point of our study is that we only genotyped 17 common pathogenic mutations and two polymorphisms of the CFTR gene, so we may have missed some rare pathogenic CFTR variants, especially in the Buryat and Yakut ethnic groups. In a clinical sense, increasing the number of rare genetic variants in a custom panel for carrier testing may reduce the ability of rapid and cost-effective identification of pathogenic variants, especially in detection of couples at risk of having a child with CF, in asymptomatic patients, newborns, or children, when the timing of diagnosis is very important. If testing using a panel of common pathogenic CFTR mutations was not successful, the next round of testing can be performed using a panel of rare pathogenic CFTR variants or NGS, which is a more expensive and complex method. It is assumed that carrier testing for rare pathogenic CFTR variants is justified in small ethnic populations with rare or unknown incidence of CF, like Buryats and Yakuts. Because targeted testing with additional rare pathogenic CFTR mutations would increase CFTR coverage, it would be desirable to perform this as a second step in defining the spectrum of pathogenic CFTR mutations in these ethnic groups.

The diversity of ethnic groups in our countries makes the development of ethno-specific CFTR mutation panels enriched for rare variants challenging. The use of NGS technology for carrier screening seems more promising in small communities with low or unknown CF prevalence than the targeted approach, since NGS can provide comprehensive testing of CFTR variants and obtain information on both known mutations and novel DNA changes in the CFTR gene with high efficiency, thereby maximizing CFTR gene coverage. In addition, NGS analysis of the CFTR gene for heterozygous carriage could be recommended at the stage of pregnancy planning, as well as for male patients with obstructive or non-obstructive forms of infertility. For example, the use of whole-exome sequencing could facilitate the analysis of both known and unknown CFTR variants and their subsequent correlation with the phenotypic features of CF and fertility parameters.

However, for clinical use, the NGS approach requires interpretation of the pathogenicity of each identified variant, since documented CFTR mutations very often have variable or unknown clinical consequences [52]. The use of NGS for carrier screening for CFTR mutations may complicate the interpretation of results in cases of variants of uncertain significance or benign variants, which are common and may lead to a significant overestimation of the frequency of pathogenic variants [68,69]. Some authors believe that genotyping of the most common pathogenic mutations is currently the most appropriate approach for routine carrier screening [49,69]. Over time, NGS methods will become more accessible for detecting rare mutations, which will allow for efficient exploration of large regions of the genome and testing for a wide range of mutations, thereby opening up prospects for the introduction of NGS technologies into routine carrier testing.

The variations in the prevalence of CFTR mutations between different ethnic groups can partly be explained by their historical background and development, based mainly on phylogenetic dissection of mtDNA and Y-chromosome haplogroups [70,71]. Specifically, Slavs of Eastern Europe are very similar in their genetic composition. Ukrainians, Belarusians and Russians have almost identical proportions of the Caucasoid and Northern European components and have virtually no Asian influence. Russians from Novosibirsk and Russian Starover exhibit ancestral proportions close to those of European Eastern Slavs; however, they also include 5–10% of Central Siberian ancestry [71]. Phylogenetic analysis of the Buryat gene pool within haplogroups N1cl and C3d revealed a strong founder effect, i.e., reduced diversity and starlike phylogeny of the median network of haplotypes that form specific subclusters. The results of a phylogenetic analysis of the haplogroups identified common genetic components for Buryats and Mongols [72]. High-resolution phylogenetic dissection of mtDNA and Y-chromosome haplogroups, as well as the analysis of autosomal SNP data, suggests that the Yakuts were colonized by repeated expansions from South Siberia [73]. The mtDNA results show a very close affinity of the Yakuts with Central Asian and South Siberian groups, which confirms their southern origin. The Y-chromosomal results confirm previous findings of a very strong bottleneck in the Yakuts, the age of which is in good accordance with the hypothesis that the Yakuts migrated north under Mongol pressure. Furthermore, the genetic results show that the Yakuts are a very homogenous population, which spread fairly recently over a very large territory [74]. Thus, based on the above considerations, the distribution patterns of CFTR mutations can be explained by historical movements of ethnic groups.

In recent decades, there has been a decline in the CF incidence, particularly in regions where prenatal or population screening for CFTR mutations has been performed [13]. The neonatal screening for pathogenic CF-causing mutations was introduced in Russia in 2006, and it also contributes to early diagnosis, timely treatment of the disease, and subsequent preservation of reproductive function, but it does not reduce the prevalence of the disease. Mandatory screening for pathogenic mutations in the CFTR gene is necessary when planning pregnancy, conducting assisted reproductive technologies, and selecting sperm or blood donors, which seems to be the most effective way to identify heterozygous carriers and reduce the burden of CF in the population. The resulting picture of ethnic differences fits into the idea that the mutation spectrum has a specific pattern for different populations, and when conducting molecular genetic diagnostics, it is necessary to take into account the ethnicity of each patient. Based on the present and similar studies, there is a need to develop ethno-specific CFTR mutation panels, especially for Asians, whose CF prevalence and genetic spectrum of CFTR mutations differ from those of Caucasians.

In Russia and Belarus, neonatal screening for CFTR mutations is performed only when clinical signs of CF or elevated levels of immunoreactive trypsinogen and a positive sweat test are detected; however, an increasing number of regions are also conducting molecular genetic testing for the most common pathogenic CFTR mutations. Due to the large number of samples requiring fast and reliable results, NGS has evidently become a first-level testing method and is likely to largely replace targeted sequencing in the near future. Moreover, if the newborn screening program is expanded to include testing for a wider spectrum of genetic diseases, NGS technology will receive significant support as the most effective way to identify carriers of pathogenic mutations.

## 4. Materials and Methods

### 4.1. Study Population

The present study was conducted on a previously collected study population of men (*n* = 1895, median age 23 years) from six cities: Arkhangelsk, Novosibirsk, Kemerovo, Ulan-Ude, Yakutsk (Russia), and Minsk (Republic of Belarus). The city of Minsk is located in the eastern part of Europe; Arkhangelsk is in the European north of Russia, within the subpolar zone; and the cities of Novosibirsk and Kemerovo are located in Western Siberia—all four cities predominantly have a Slavic population (approximately 90–95%). The cities of Ulan-Ude and Yakutsk are located in Eastern Siberia, with Yakutsk being close to the Arctic. Buryats make up 32% of the total population of Ulan-Ude, while Yakuts account for 43% of the total population of Yakutsk. The geographical location of these cities is shown in Appendix A.

In all cities, the study design and standardized sampling protocol were the same, as previously described in detail [40,75]. Briefly, this study involved male volunteers, apparently unrelated, from the general population, regardless of their fertility status. All participants were either born or had lived for at least 3–5 years in the cities where this study was conducted. The overwhelming majority of men at the time of examination were employees or students of higher education institutions and had not previously undergone andrological examinations. Inclusion criteria for participation in this study included the absence of acute general illnesses or chronic diseases in an acute phase, as well as urogenital infections. Each participant completed a standardized questionnaire that included information about age, place of birth, nationality, marital status, and certain lifestyle characteristics. The data of the participants were kept anonymous. Ethnic background was assessed for up to three generations—for the participant, his parents, and maternal and paternal grandparents. All men included in this study gave informed consent to participate in the examination. The Ethics Committee of the Federal Research Center “Institute of Cytology and Genetics” of the Siberian Branch of the Russian Academy of Sciences approved this study (protocol No. 160 dated 17 September 2020).

The study population was multinational and consisted of men from over 20 different nationalities, including descendants of ethnically mixed marriages. The investigation of ethnic differences in the spectrum and carrier frequency of pathogenic variants of the CFTR gene, as well as the possible influences of different gene variants on semen parameters and hormonal profiles, was conducted on the three most numerous ethnic groups: Slavs (Russians, Belarusians, and Ukrainians), Buryats, and Yakuts. The group of Slavs consisted of men who lived in all six cities, and the group of Buryats or Yakuts included men from the cities of Ulan-Ude and Yakutsk, respectively.

### 4.2. Physical Examination, Blood and Semen Collection, and Semen and Hormonal Analysis

In each city, all participants were examined by the same experienced andrologist, and the results of the examination, particularly the current urogenital disorders, were recorded. None of the participants had any clinical manifestations or family history indicating CF or CBAVD. The age of all participants was noted, body weight and height were measured, and the testicular volume was estimated by the Prader orchidometer and was presented as bitesticular volume (paired testicular volume). A fasting blood sample was taken from the cubital vein in the morning (before the semen collection) from each participant. Serum samples were stored at approximately 40 °C until hormonal analysis was conducted. Semen samples were collected by masturbation into disposable sterile plastic containers in a special private room. Each participant was asked in advance about the need for sexual abstinence at least 2–3 days before the examination; the abstinence time was recorded according to the information given by the participant.

The semen samples were analyzed for semen volume (mL), sperm concentration (× 10^6^/mL), sperm motility (categories A + B), and normal morphology (percentage) according to the WHO laboratory manual for the examination and processing of human semen [38,39]. Semen analysis was described earlier elsewhere in more detail [40,75]. Sperm concentration was assessed using Goryaev’s hemocytometer under light microscope (magnification ×400). Total sperm count was then calculated by multiplying the individual’s sperm concentration by the semen volume. Percentage of spermatozoa with progressive motility (categories A + B) was estimated in native ejaculate using an automatic sperm analyzer SFA-500 (Biola, Moscow, Russia). To assess sperm morphology, ejaculate smears were prepared, fixed with methanol, and stained using commercially available kits, Diff-Quick (Abris plus, St Petersburg, Russia), according to the manufacturer’s manual. Two hundred spermatozoa were examined for morphology with an optical microscope (Axio Skop.A1, Carl Zeiss, Oberkochen, Germany) at ×1000 magnification with oil immersion, and the sperm morphological anomalies were listed according to the WHO guidelines [39]. Sperm morphology evaluations were carried out in duplicates in random and blinded order, and we report here the percentage of sperm scored as morphologically normal (percentage).

Serum hormone concentrations were determined by enzyme immunoassay using commercially available kits—«Steroid IFA-Testosterone-01», «Gonadotropin IFA-LH», «Gonadotropin IFA-FSH» (Alkor Bio, St Petersburg, Russia), «Estradiol-IFA» (Xema Medica, Russia), and «Inhibin B Gen II ELISA» (Beckman Coulter, Brea, CA, USA)—according to the manufacturer’s manuals. The ranges of evaluated concentrations for total testosterone (T), estradiol (E_2_), follicle stimulating hormone (FSH), luteinizing hormone (LH), and inhibin B (InhB) were 0.2–50 nmol/L, 0.1–20 nmol/L, 2.0–100 mME/mL, 20–90 mME/mL, and 12–105 pg/mL, respectively. The sensitivities for T, E_2_, FSH, LH, and InhB were 0.2 mmol/L, 0.025 nmol/L, 0.25 mME/mL, 0.25 mME/mL, 2.6 pg/mL, respectively. The intra- and interassay coefficients of variation were as follows: T < 8.0%, E_2_ < 8.0%, FSH < 8.0%, LH < 8.0%, and InhB ≤ 6.8%.

### 4.3. Genetic Testing

Genomic DNA from peripheral blood leukocytes was extracted by the conventional phenol-chloroform method [76]. This study analyzed 17 mutations and 2 polymorphisms of the CFTR gene classified as pathogenic: (1) G542X, (2) F508del, (3) I507del, (4) E92K, (5) E92X, (6) L138ins, (7) W1282X, (8) N1303K, (9) 2184insA, (10) 3849+10kb C>T, (11) 2143delT, (12) 1677delTA, (13) R334W, (14) S1196X, (15) R117C, (16) CFTRdele2,3(21kb), (17) L467F, (18) IVS9-5T, and (19) 5T (12-13) TG.

The CFTR variants, which were used in our study population, were selected taking into account their high prevalence and pathogenicity. The majority of them were also drawn from the CF registers “The Clinical and Functional Translation of CFTR (CFTR2)” [17] and “The Register of Patients with Cystic Fibrosis in the Russian Federation. 2023.” [14], as well as the published data on spectrum and allele frequency of the CFTR variants in CF or CBAVD syndrome patients in the Russian Federation [18,20,30,32,37,77]. As a result, 14 CFTR CF-causing variants with high allele frequencies in CF patients were selected (Appendix A). In addition, we selected two rare variants, R117C and E92X, which were identified in CF or CBAVD syndrome patients in Russia [18,20]. Finally, the L467F variant was added because it forms a complex allele with the F508del mutation and leads to ineffectiveness of targeted therapy [78]. According to “The Register of Patients with Cystic Fibrosis in the Russian Federation. 2023.” [14], the CFTR coverage of the selected 17 CF-causing variants is 78.2% of the total frequency of pathogenic variants in Russian CF patients. Annotation of CFTR variants included in this study is given in the Appendix A. The trivial (traditional) gene names are used in this paper. Polymorphisms of the IVS9-5T alone or in the *cis*-combination with the 5T-12TG or 5T-13TG in the CFTR gene were selected because they can cause splicing disorders and various clinical consequences, including impaired spermatogenesis in other populations [27,59,62].

The identification of CFTR mutations was performed using real-time PCR, which employed specific probes for the studied DNA regions, having previously simultaneously analyzed two mutations of the CFTR gene in one tube. The method was described earlier [79]. The TG repeats were analyzed only in those samples where the 5T allele was present. The number of T repeats was determined in a single tube using three probes that provided signals in different channels. To detect the extensive deletion of 2,3 exons in all samples and to assess the number of (TG)n repeats in carriers of the 5T allele, a PCR method based on TaqMan technology was utilized. The *cis* or *trans* position was determined by sequencing; for this purpose, samples identified with the combinations of 5T and 12TG or 5T and 13TG were sequenced using forward and reverse primers. Sequencing to confirm the nucleotide sequence of the identified mutations was carried out using the Sanger method. The oligonucleotides for the studied regions of the CFTR gene are shown in Appendix A.

The freeze-dried reaction mixture (Vector-Best, Novosibirsk, Russia) contained all the necessary components for PCR: 67 mM Tris-HCl (pH 8.9), 50 mM KCl, 17 mM (NH_4_)_2_SO_4_, 0.1% Tween-20, 5 mM MgCl_2_, 0.4 mM of each of the four deoxynucleoside triphosphates, 0.1 mg/mL bovine serum albumin, 1 unit of Taq polymerase in combination with antibodies to its active site (Clontech, Mountain View, CA, USA), 0.5 µM of primers with a fluorescence quencher, 0.125 µM of unlabelled primers, 0.25 µM of probes, and 0.25 µM of blocked oligonucleotides. A sample of the analyzed DNA was added in a volume of 50 µL.

The amplification reaction and detection were performed on a thermal cycler with an optical module CFX96 Touch Real-Time PCR Detection System (Bio-Rad, Hercules, CA, USA). The thermal cycling conditions were as follows: 1 cycle—2 min at 50 °C 1; 1 cycle—2 min at 94 °C; 50 cycles—10 s at 94 °C, 20 s at 60 °C; and melting–temperature change from 27 to 80 °C in 1 °C increments, with incubation for 5 s at each step and fluorescence registration.

For validation, we used sequencing of the studied CFTR gene regions containing mutations. We attach tables with primers and probes that we used in our mutation detection method (Appendix A). All probes are designed in such a way that they are completely complementary to the sequences containing the mutant allele of CFTR. In the case of complete complementarity with mutant allele, we observe a peak at high temperature in PCR, and in the case of a normal allele, a peak at low temperature (Appendix A). The method was validated using Sanger sequencing (Appendix A).

### 4.4. Statistical Analysis

Statistical data analysis was performed using the STATISTICA software package (version 8.0). The carrier frequencies of pathogenic CFTR variants were compared between different ethnic groups or groups with different semen quality (azoospermia, pathozoospermia, and normozoospermia) using the Yates’ Chi-squared (χ^2^) test. To investigate the effect of the imbalance in the number of ethnic groups in the contingency table under verification of interethnic differences in the carrier frequency of pathogenic CFTR variants, we carried out simulation study of contingency tables under smaller and equal number ethnic groups as for Buryat (*n* = 223) and Yakut (*n* = 147) groups. It turned out that all the main statistical patterns remained the same as in the analysis of the original contingency table. Only a decrease in the number of cases in each group to 65 led to statistically insignificant interethnic differences. Thus, the analysis of the original contingency table does indeed reflect the identified ethnic differences in the carrier frequency of pathogenic CFTR variants. One-way Kruskal–Wallis ANOVA was performed to identify differences in anthropometric, semen, and hormonal parameters between groups of carriers and non-carriers of CFTR mutations. Duncan’s test was applied for multiple comparisons of groups. Differences were considered statistically significant when *p* < 0.05.

## 5. Conclusions

In men from the general population (*n* = 1895), the carrier frequencies of the most common 17 pathogenic mutations of the CFTR gene associated with CF were studied. A total of 39 heterozygous carriers of CFTR mutations (2.06%), 104 carriers of the IVS9-5T polymorphism (5.49%), and 21 carriers of the 5T-12/13TG variant (1.11%) were identified. The most common mutation was F508del, with a carrier frequency of 1.53%.

When comparing the three most numerous groups, Slavic, Buryat, and Yakut, ethnic differences in the spectrum and carrier frequency of pathogenic CFTR variants were identified. Heterozygous carriers of CFTR gene mutations (2.70%) were found only among Slavs; none were detected among Buryats and Yakuts. The highest carrier frequency of CFTR polymorphisms was observed among Slavs (8.35%), followed by Buryats (5.83%), and the lowest value was among Yakuts (1.36%). The data obtained confirm the need to develop population-specific CFTR mutation panels.

No significant differences were found in the carrier frequency of pathogenic CFTR gene variants among men with azoospermia, pathozoospermia, and normozoospermia, as well as in semen parameters and hormonal profiles among carriers and non-carriers of pathogenic CFTR gene variants in Slavs and Buryats. The absence of negative effects of carriage of pathogenic CFTR variants on male fertility indicators could contribute to their widespread distribution and the persistent prevalence of cystic fibrosis and infertility within the general population.

## Figures and Tables

**Table 1 ijms-26-06625-t001:** Spectrum and carrier frequency of identified pathogenic CFTR variants among men of the entire study population and selected ethnic groups (carriers’ number/carrier frequency/95% CI).

CFTR Genotype	Entire Study Population, *n* = 1895	Slavs, *n* = 1186	Buryats, *n* = 223	Yakuts, *n* = 147
F508del/N	29/0.0153 (0.0098–0.0208)	24/0.0202 (0.0122–0.0282)	0/0.0000 (0.0000–0.0000)	0/0.0000 (0.0000–0.0000)
G542X/N	1/0.0005 (0.0000–0.0016)	0/0.0000 (0.0000–0.0000)	0/0.0000 (0.0000–0.0000)	0/0.0000 (0.0000–0.0000)
N1303K/N	2/0.0011 (0.0000–0.0025)	2/0.0017 (0.0000–0.0040)	0/0.0000 (0.0000–0.0000)	0/0.0000 (0.0000–0.0000)
3849+10kbC>T/N	2/0.0011 (0.0000–0.0025)	1/0.0008 (0.0000–0.0025)	0/0.0000 (0.0000–0.0000)	0/0.0000 0.0000–0.0000
CFTRdele2,3/N	1/0.0005 (0.0000–0.0016)	1/0.0008 (0.0000–0.0025)	0/0.0000 (0.0000–0.0000)	0/0.0000 (0.0000–0.0000)
R117C/N	1/0.0005 (0.0000–0.0016)	1/0.0008 (0.0000–0.0025)	0/0.0000 (0.0000–0.0000)	0/0.0000 (0.0000–0.0000)
F508del/5T	1/0.0005 (0.0000–0.0016)	1/0.0008 (0.0000–0.0025)	0/0.0000 (0.0000–0.0000)	0/0.0000 (0.0000–0.0000)
F508del/5T-12TG	1/0.0005 (0.0000–0.0016)	1/0.0008 (0.0000–0.0025)	0/0.0000 (0.0000–0.000)0	0/0.0000 (0.0000–0.0000)
R117C/5T	1/0.0005 (0.0000–0.0016)	1/0.0008 (0.0000–0.0025)	0/0.0000 (0.0000–0.0000)	0/0.0000 (0.0000–0.0000)
**Total heterozygotes**	**39/0.0206** **(0.0142–0.0270)**	**32/0.0270** **(0.0178–0.0362)**	**0/0.0000** **(0.0000–0.0000)**	**0/0.0000** **(0.0000–0.0000)**
IVS9-5T/N	104/0.0549 (0.0446–0.0651)	90/0.0759 (0.0608–0.0910)	3/0.0135 (0.0000–0.0286)	1/0.0068 (0.0000–0.0201)
5T/5T	1/0.0005 (0.0000–0.0016)	1/0.0008 (0.0000–0.0025)	0/0.0000 (0.0000–0.0000)	0/0.0000 (0.0000–0.0000)
5T-12TG/N	11/0.0058 (0.0024–0.0092)	8/0.0047 (0.0021–0.0114)	0/0.0000 (0.0000–0.0000)	1/0.0068 (0.0000–0.0201)
5T-13TG/N	10/0.0053 (0.0020–0.0085)	0/0.0000 (0.0000–0.0000)	10/0.0448 (0.0177–0.0720)	0/0.0000 (0.0000–0.0000)
**Total polymorphisms**	**126/0.0665** **(0.0553–0.0777)**	**99/0.0835** **(0.0677–0.0992)**	**13/0.0583** **(0.0275–0.0890)**	**2/0.0136** **(0.0000–0.0323)**

**Table 2 ijms-26-06625-t002:** Spectrum and carrier frequency of identified pathogenic CFTR variants among men with normal and impaired spermatogenesis (carriers’ number/carrier frequency/95%CI).

CFTR Genotype	Normozoospermia *n* = 1119	Pathozoospermia *n* = 675	Azoospermia *n* = 37
F508del/N	19/0.0170 (0.0094–0.0245)	8/0.0119 (0.0037–0.0200)	0/0.0000 (0.0000–0.0000)
G542X/N	0/0.0000 (0.0000–0.0000)	1/0.0015 (0.0000–0.0044)	0/0.0000 (0.0000–0.0000)
N1303K/N	1/0.0009 (0.0000–0.0026)	1/0.0015 (0.0000–0.0044)	0/0.0000 (0.0000–0.0000)
3849+10kbC>T/N	2/0.0018 (0.0000–0.0043)	0/0.0000 (0.0000–0.0000)	0/0.0000 (0.0000–0.0000)
CFTRdele2,3/N	1/0.0009 (0.0000–0.0026)	0/0.0000 (0.0000–0.0000)	0/0.0000 (0.0000–0.0000)
R117C/N	1/0.0009 (0.0000–0.0026)	0/0.0000 (0.0000–0.0000)	0/0.0000 (0.0000–0.0000)
F508del/5T	1/0.0009 (0.0000–0.0026)	0/0.0000 (0.0000–0.0000)	0/0.0000 (0.0000–0.0000)
F508del/5T-12TG	1/0.0009 (0.0000–0.0026)	0/0.0000 (0.0000–0.0000)	0/0.0000 (0.0000–0.0000)
R117C/5T	1/0.0009 (0.0000–0.0026)	0/0.0000 (0.0000–0.0000)	0/0.0000 (0.0000–0.0000)
**Total heterozygotes**	**27/0.0241** **(0.0151–0.0331)**	**10/0.0148** **(0.0057–0.0239)**	**0/0.0000** **(0.0000–0.0000)**
IVS9-5T/N	58/0.0518 (0.0388–0.0648)	42/0.0622 (0.0440–0.0804)	1/0.0270 (0.0000–0.0793)
5T/5T	1/0.0009 (0.0000–0.0026)	0/0.0000 (0.0000–0.0000)	0/0.0000 (0.0000–0.0000)
5T-12TG/N	6/0.0054 (0.0011–0.0096)	6/0.0089 (0.0018–0.0160)	0/0.0000 (0.0000–0.0000)
5T-13TG/N	8/0.0071 (0.0022–0.0121)	1/0.0015 (0.0000–0.0044)	0/0.0000 (0.0000–0.0000)
**Total polymorphisms**	**73/0.0652** **(0.0508–0.0797)**	**49/0.0726** **(0.0530–0.0922)**	**1/0.0270** **(0.0000–0.0793)**

Note: Normozoospermia—sperm concentration ≥ 16.0 mill/mL, proportion of progressive motile sperm ≥ 30%, proportion of morphologically normal sperm ≥ 4.0% [38,39]; Pathozoospermia—concentration and proportion of progressively motile and morphologically normal sperm are below the reference values (either each indicator or any combination thereof); Azoospermia—no sperm in the ejaculate.

**Table 3 ijms-26-06625-t003:** Anthropometric, semen, and hormonal parameters in carriers and non-carriers (Control) of CFTR gene mutations in Slavic and Buryat groups (mean ± SEM).

Parameter	Slavs, *n* = 1186				Buryats, *n* = 206	
	Control *n* = 1055	Heterozygotes *n* = 29	IVS9-5T *n* = 90	5T-12TG *n* = 8	Control *n* = 196	5T-TG13 *n* = 10
Age (years)	24.8 ± 0.2	25.0 ± 1.5	25.7 ± 0.7	27.6 ± 2.1	23.9 ± 0.5	23.3 ± 1.7
Weight (kg)	78.4 ± 0.4	73.1 ± 1.7	79.6 ± 1.4	78.1 ± 3.5	70.7 ± 0.9	66.3 ± 2.9
Height (cm)	179.4 ± 0.2	178.4 ± 1.1	179.2 ± 0.8	178.5 ± 2.5	175.0 ± 0.4	173.2 ± 1.6
BTV (mL)	42.5 ± 0.3	42.9 ± 1.3	42.6 ± 0.9	43.5 ± 2.8	35.6 ± 0.5	34.5 ± 2.5
Semen volume (mL)	3.7 ± 0.1	3.7 ± 0.4	3.9 ± 0.2	3.5 ± 0.4	3.2 ± 0.1	2.9 ± 0.4
TSC (10^6^/ejaculate)	242.4 ± 7.3	217.2 ± 36.6	201.5 ± 17.7	263.9 ± 130.5	135.2 ± 8.6	134.3 ± 23.2
Sperm concentration (10^6^/mL)	66.35 ± 1.68	59.11 ± 7.77	54.10 ± 4.67	61.44 ± 23.43	44.49 ± 2.62	48.01 ± 6.96
Sperm motility (%)	48.0 ± 0.8	44.8 ± 4.6	43.1 ± 3.0	34.6 ± 12.0	45.7 ± 2.0	55.8 ± 8.8
Normal morphology (%)	6.96 ± 0.10	6.40 ± 0.49	6.67 ± 0.37	5.08 ± 1.22	6.79 ± 0.205	7.69 ± 0.48
LH (IU/L)	3.24 ± 0.04	3.31 ± 0.31	3.37 ± 0.13	2.89 ± 0.46	4.03 ± 0.12	3.21 ± 0.25
FSH (IU/L)	3.68 ± 0.09	3.17 ± 0.22	3.81 ± 0.23	2.73 ± 0.39	4.73 ± 0.21	3.50 ± 0.55
Testosterone (nmol/L)	22.81 ± 0.25	22.94 ± 1.29	22.36 ± 0.77	20.77 ± 2.54	18.83 ± 0.438	17.00 ± 1.92
Estradiol (nmol/L)	0.207 ± 0.002	0.196 ± 0.011	0.200 ± 0.009	0.169 ± 0.018	0.234 ± 0.008	0.237 ± 0.017
Inhibin B (pg/mL)	188.9 ± 2.1	188.0 ± 10.8	190.3 ± 7.4	217.6 ± 24.47	146.2 ± 4.5	168.7 ± 10.8

Note: BTV—bitesticular volume (paired testicular volume); TSC—total sperm count in an ejaculate; sperm motility—a percentage of motile sperm; Normal morphology—a percentage of morphologically normal sperm; LH—luteinizing hormone; FSH—follicle stimulating hormone; heterozygotes are heterozygous carriers of the following pathogenic variants: F508del, G542X, N1303K, 3849+10kbC>T, CFTRdele2,3, and R117C.

## Data Availability

Restrictions apply to the availability of some data of this study to preserve patient confidentiality. The corresponding author will on request detail the restrictions and other conditions under which access to some data may be provided.

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
