# Peer review of "The Spectrum and Carrier Frequencies of Common Pathogenic Cystic Fibrosis Transmembrane Conductance Regulator Gene Mutations in Men from the General Population: The Role of Ethnicity"

_ijms, 2025, doi:10.3390/ijms26146625_

Round 1

Reviewer 1 Report

Comments and Suggestions for Authors

The manuscript entitled ‘the spectrum and carriage frequencies of pathogenic CFTR gene mutations in men from the general population: a role of ethnicity’ is well written by need several modifications for publication.

Authors tested a very restricted limited number of mutations (17 CFTR mutations), while 264 different pathogenic variants were recorded in Russia. Consequently the manuscript cannot be defined as a carrier frequency spectrum. The title must be changed, and the manuscript structure also.

Explain why you tested only those mutations.

Substitute ‘carriage’ with ‘carrier’.

Part 4.3: eliminate AZF test as it is not related with the present study.

Reviewer 2 Report

Comments and Suggestions for Authors

Dear Dr. Osadchuk and colleagues,

Thank you for submitting your manuscript examining CFTR gene mutations in a multi-ethnic population from Russia and Belarus. This study addresses an important gap in understanding ethnic variations in CFTR mutation frequencies and their clinical implications. While the work presents valuable data, several significant concerns require attention before publication.

Major Revisions Required

  1. Statistical Analysis and Sample Size Limitations

The statistical approach requires substantial improvement. The Yakut group (n=147) and Buryat group (n=223) are considerably smaller than the Slavic cohort (n=1,186), creating significant power imbalances that may lead to false negative findings rather than true absence of mutations. You should conduct formal power calculations to determine the minimum sample sizes required to detect clinically meaningful mutation frequencies. Additionally, consider applying correction methods for multiple comparisons when analyzing ethnic differences.

  1. Genetic Analysis Scope and Methodology

The limitation to 17 pathogenic mutations represents a critical weakness that undermines your conclusions about ethnic-specific mutation spectra. Modern approaches would employ next-generation sequencing or comprehensive CFTR gene panels to capture the full mutation landscape. You must acknowledge this limitation more prominently and discuss how it affects the interpretability of your negative findings in Asian populations.

  1. Population Stratification and Ethnic Classification

The ethnic classification methodology lacks sufficient rigor. You mention assessment "for up to two generations" but provide no details about handling mixed ancestry or validation of self-reported ethnicity. Consider incorporating genetic ancestry markers or principal component analysis to confirm population structure, particularly given the complex demographic history of Siberian populations.

  1. Clinical Correlation and Functional Assessment

The absence of correlation between CFTR mutations and fertility parameters requires more sophisticated analysis. Consider stratifying by mutation severity classes, evaluating dose-response relationships, and including additional reproductive endpoints such as sperm DNA fragmentation or oxidative stress markers that may be more sensitive to CFTR dysfunction.

Minor Revisions Required

  1. Methodological Clarity

Provide more detailed information about the real-time PCR methodology, including primer sequences, probe specifications, and validation procedures. The supplementary tables are referenced but the actual methodology description remains insufficient for reproducibility.

  1. Data Presentation and Statistical Reporting

Several statistical results lack appropriate confidence intervals and effect sizes. Table formatting should be improved for clarity, and consider presenting mutation frequencies with 95% confidence intervals. The chi-square test results should include exact p-values rather than threshold reporting.

  1. Discussion Enhancement

The discussion would benefit from deeper exploration of the evolutionary and demographic factors that may explain the observed ethnic differences. Consider discussing founder effects, genetic drift, and historical population movements that could account for the mutation distribution patterns.

  1. Literature Integration

Expand the comparison with international studies, particularly those from other Asian populations. The current literature review focuses heavily on Russian studies and would benefit from broader geographic representation.

  1. Clinical Implications

Strengthen the discussion of clinical implications for genetic counseling and carrier screening programs. Provide specific recommendations for implementation of ethnicity-specific screening protocols.

I recommend acceptance pending these revisions, as the study addresses clinically relevant questions and provides valuable population-specific data. However, the methodological limitations and analytical concerns must be adequately addressed to ensure the reliability and clinical utility of your findings.

Sincerely,

Reviewer

Comments on the Quality of English Language

The English could be improved to more clearly express the research.

Round 2

Reviewer 1 Report

Comments and Suggestions for Authors

The manuscript entitled ‘the spectrum and carriage frequencies of pathogenic CFTR gene mutations in men from the general population: a role of ethnicity’ is well written by need several modifications for publication.

Authors tested a very restricted limited number of mutations (17 CFTR mutations), while 264 different pathogenic variants were recorded in Russia. Consequently the manuscript cannot be defined as a carrier frequency spectrum. The title must be changed, and the manuscript structure also.

Explain why you tested only those mutations.

Substitute ‘carriage’ with ‘carrier’.

Part 4.3: eliminate AZF test as it is not related with the present study.

Reviewer 2 Report

Comments and Suggestions for Authors

Dear Dr. Osadchuk,

Thank you for addressing the previous review comments comprehensively. After carefully reviewing your revised manuscript, I have a few minor suggestions to further enhance the clarity and impact of your research:

  1. Methodology Clarification
  • While the statistical approach has been improved, consider adding a brief paragraph in the methods section explicitly detailing the power calculations and multiple comparison correction methods used.
  1. Ethnic Classification
  • The explanation of ethnic background assessment is helpful. However, consider adding a limitation statement acknowledging potential constraints of self-reported ethnicity and the need for future genetic ancestry marker validation.
  1. Clinical Implications
  • Your discussion of potential screening strategies is valuable. Consider adding a concise decision-making flowchart or algorithm illustrating the recommended approach for carrier screening across different ethnic groups.
  1. Future Research Directions
  • The manuscript would benefit from a more explicit outline of future research needs, particularly regarding Next-Generation Sequencing (NGS) implementation in diverse populations.
  1. Supplementary Data
  • Consider including a supplementary table mapping the geographic distribution of the studied populations to provide additional context.

These suggestions are intended to further strengthen an already robust manuscript. The revisions are minor and should not require substantial rewriting.

Sincerely,

Reviewer
